# Aggression, Impulsivity and Suicidal Behavior in Depressive Disorders: A Comparison Study between New York City (US), Madrid (Spain) and Florence (Italy)

**DOI:** 10.3390/jcm10143057

**Published:** 2021-07-09

**Authors:** Javier-David Lopez-Morinigo, Maura Boldrini, Valdo Ricca, Maria A. Oquendo, Enrique Baca-García

**Affiliations:** 1Department of Psychiatry, Universidad Autónoma de Madrid, 28049 Madrid, Spain; ebacgar2@yahoo.es; 2Department of Psychiatry, IIS-Fundación Jiménez Díaz, 28040 Madrid, Spain; 3Centro de Investigación Biomédica en Red Salud Mental (CIBERSAM), 28029 Madrid, Spain; 4Department of Child and Adolescent Psychiatry, Institute of Psychiatry and Mental Health, Hospital General Universitario Gregorio Marañón, IiSGM, CIBERSAM, School of Medicine, Universidad Complutense, 28009 Madrid, Spain; 5Department of Psychiatry, New York State Psychiatric Institute, Columbia University Irvin Medical Center, New York, NY 10032, USA; mb928@cumc.columbia.edu; 6Department of Health Sciences, University of Florence, 50121 Florence, Italy; valdo.ricca@unifi.it; 7Department of Psychiatry, Perelman School of Medicine, University of Pennsylvania, Philadelphia, PA 19104, USA; moquendo@mail.med.upenn.edu; 8Department of Psychiatry, University Hospital Rey Juan Carlos, 28933 Mostoles, Spain; 9Department of Psychiatry, General Hospital of Villalba, 28400 Madrid, Spain; 10Department of Psychiatry, University Hospital Infanta Elena, 28342 Valdemoro, Spain; 11Universidad Católica del Maule, Talca 3466706, Chile; 12Department of Psychiatry, Centre Hospitalier Universitaire de Nîmes, 30900 Nîmes, France

**Keywords:** aggression, impulsivity, suicidal behavior

## Abstract

The association of aggression and impulsivity with suicidal behavior (SB) in depression may vary across countries. This study aimed (i) to compare aggression and impulsivity levels, measured with the Brown-Goodwin Scale (BGS) and the Barratt Impulsivity Scale (BIS), respectively, between New York City (NYC) (US), Madrid (Spain) and Florence (Italy) (ANOVA); and (ii) to investigate between-site differences in the association of aggression and impulsivity with previous SB (binary logistic regression). Aggression scores were higher in NYC, followed by Florence and Madrid. Impulsivity levels were higher in Florence than in Madrid or NYC. Aggression and impulsivity scores were higher in suicide attempters than in non-attempters in NYC and in Madrid. SB was associated with aggression in NYC (OR 1.12, 95% CI 1.07–1.16; *p* < 0.001) and in Florence (OR 1.11, 95% CI 1.01–1.22; *p* = 0.032). Impulsivity was linked with SB in NYC (OR 1.01, 95% CI 1.00–1.02; *p* < 0.001) and in Madrid (OR 1.03, 95% CI 1.02–1.05; *p* < 0.001). The higher suicide rates in NYC, compared to Madrid or Florence, may be, in part, explained by these cross-cultural differences in the contribution of aggression-impulsivity to SB, which should be considered by future research on SB prevention.

## 1. Introduction

Suicide has undoubtedly become a major public health issue, with almost one million deaths every year across the world [1]. Of concern, is the fact that suicide rates are likely to significantly increase worldwide due to the COVID-19-related economic turndown [2,3,4,5], consistent with previous economic recessions [6]. In the US over 48,000 people took their lives in 2018, which represents an overall increase in approximately 35% in comparison with 1999 [7]. According to psychological autopsy studies, up to 74% of those who ended their lives met criteria for depression [8]. Consistent with the stress-diathesis model of suicide, aggression and impulsivity have been established as risk factors for suicidal behavior (SB) [9,10,11], particularly in unipolar [12] and bipolar depression [13]. Interestingly, aggression and impulsivity were reported to share neurobiological mechanisms with SB, particularly decreased serotonergic activity [9,14,15,16], which may therefore become a potential pharmacological target for the prevention and management of SB. However, a large cohort of depressed patients from France (*n* = 4041) found suicidal ideation and previous SB to predict poor response to antidepressants in terms of achieving depression remission independently of the type of antidepressant and other comorbidities [17].

Two previous studies from our group, which compared two samples of suicide attempters with non-attempters’ major depressive disorder (MDD) patients and healthy controls from New York City (NYC) and Madrid (Spain) [18,19], showed that the greater lethality of SB in NYC subjects was related to their higher aggression levels in comparison with Madrid participants, while no differences in impulsivity scores between suicide attempters and healthy controls were revealed by the analyses. This said, it is worth noting that these studies tested different hypotheses. First, the role of the promoter area of the serotonin transporter (17q11.1-q12) was investigated in relation to history of aggression and impulsivity in two samples of suicide attempters and healthy controls from NYC and Madrid, which partially overlapped with the samples used in the present study [19]. The second study aimed to compare the association between lethality of suicide attempts and aggression levels in Madrid and in NYC with the aforementioned two samples [18]. However, the present study aimed to compare aggression and impulsivity levels across three sites, namely NYC, Madrid and Florence, and the relationship between both variables and SB in three samples of “depressive disorders patients”, that is, we did not include healthy controls. Hence, the present study is not a partial or full re-analysis of previously published data. More specifically, unlike the aforementioned two reports [18,19], not only new data were collected in NYC and in Madrid, but also we included a third cohort of depressive disorders patients from Florence in order to test this study’s hypotheses. However, as acknowledged in the methods and limitations sections, these three samples were not originally recruited for this study. Rather, based on an international collaboration research network, the three groups decided to share data, thus allowing this comparison study (also, see recruitment dates below).

By building on this work, we have conducted the present study aimed: (i) to compare aggression and impulsivity levels across three major cities in different countries, namely New York City (NYC), Madrid (Spain) and Florence (Italy) and (ii) to investigate between-site differences in the association of aggression and impulsivity with previous SB. In particular, the following hypotheses were to be tested: (i) that NYC subjects will show higher aggression and impulsivity levels than those from Madrid and Florence; (ii) that suicide attempters will have higher aggression and impulsivity levels than non-attempters across sites, (iii) that while both aggression and impulsivity will be significantly associated with previous SB in NYC, these associations will not reach significance in Madrid or in Florence. In particular, hypothesis i and iii were based on our aforementioned previous studies [18,19], which are supported further by previous epidemiological research on suicide rates differences across countries [20]. The association of aggression-impulsivity with SB (hypothesis ii) has been previously reported [9,10,11].

## 2. Materials and Methods

### 2.1. Participants

Three samples of depressed patients from New-York City (NYC) (*n* = 647), Madrid (Spain) (*n* = 658) and Florence (Italy) (*n* = 93) were included in this study.

In NYC, suicide attempters were recruited from a population of depressed patients at a university psychiatric hospital from 1998 to 2001.

In Madrid, depressed suicide attempters from the emergency department (ED) of a publicly-funded university hospital (Ramón y Cajal University Hospital), which provides care to a population of approximately 500,000 inhabitants residing in North-Madrid, were recruited over one year (1999). This hospital covers all medical emergencies in this catchment area with the only exception for those presenting to local private hospitals, which are likely to be a tinny proportion of ED users. Since there are no psychiatric hospitals in this area, the vast majority of suicide attempters are referred to this hospital so they can be admitted to the psychiatric ward as appropriate.

In Florence participants were recruited from Careggi University Hospital, which is the main hospital of the city of Florence, with 1322 beds and approximately 50,000 admissions per year, over 2000–2005. Depressed suicide attempters were hospitalized in the Inpatient Unit of the Department of Psychiatry.

We applied the same inclusion/exclusion criteria across sites. In particular, those adults—age 18–64, both inclusive—who made a suicide attempt (suicide attempters) as defined by O’Carroll and colleagues, i.e., “any act the patient deliberately performed to kill himself or herself” [21] were administered either the Structured Clinical Interview (SCID) [22] (NYC) or the Mini-International Neuropsychiatric Interview [23] (Madrid and Florence samples), that is, a brief structured interview which made the major Axis I DSM-IV diagnosis. Those who were found to meet criteria for Major Depressive Disorder (MDD) or Bipolar Depression (BD) were invited to participate in this study and those who agreed provided written informed consent. Those with a primary diagnosis of schizophrenia, eating disorder, alcohol or substance use disorders, Obsessive-Compulsive Disorder (OCD) and/or suffering a neurological condition were excluded. Of note, we considered “Bipolar” depression an inclusion criterion given its well-recognized association with SB, aggression and impulsivity [13]. Suicide non-attempters were recruited from these sites following the same procedure. Thus, non-matched adults (age 18–64 years, both inclusive) who presented with MDD or BD according to the SCID [22] (NYC) or MINI [23] (Madrid and Florence) to these three hospitals over the aforementioned periods and met no exclusion criteria were invited to participate in the study.

Participants provided written informed consent as approved by the local institutional review boards.

### 2.2. Measures

Aggression and impulsivity were the co-primary outcome measures and assessed using the same instruments across sites. Aggression history was rated using the Brown-Goodwin Scale (BGS) [24], which measures lifetime aggressive acts with 10 items, each of which ranges from 1 to 4, hence total scores range from 10 to 40. An 11th item measures suicidal behavior and, therefore, is excluded. The original version of the scale was used since no language-related issues were raised to rate lifetime aggressive acts [24]. Impulsivity was measured with the Barratt Impulsiveness Scale (BIS) [25], which assesses impulsivity traits. The 30-item version of the scale, which provides total scores ranging from 0 to 120, was used. The Spanish [26] and Italian [27] versions of the BIS were used in Madrid and Florence, respectively. We considered BGS and BIS total scores for the analyses.

Additional variables included: age at study enrolment, gender, ethnicity, marital status, education level, as well as alcohol and substances use disorders, based on the Structured Clinical Interview (SCID) [22] for DSM-IV in NYC, and on the Mini-International Neuropsychiatric Interview [23] in Madrid and Florence. We decided to consider comorbid alcohol and substance use disorders given their well-recognized association with SB, aggression and impulsivity [28].

With regard to SB, as alluded to above, we used the same O’Carroll et al.’s SB consensus definition across sites, that is, “any act the patient deliberately performed to kill himself or herself” [21].

### 2.3. Statistics

First, in order to test hypothesis i, i.e., between-site differences in aggression and impulsivity levels, BGS and BIS total scores were compared across sites through an analysis of variance model (ANOVA). We further adjusted the ANOVA models for age and gender as co-variables (ANCOVA), which did not alter the results below. Second, regarding hypotheses ii, that is, differences in aggression and impulsivity levels between suicide attempters and non-attempters, BGS and BIS total scores means were compared between groups (suicide attempters vs. non-attempters), respectively, for each site since distributions were sufficiently normal. Finally, binary logistic regression analyses provided univariate Odds Ratios (OR), including 95% confidence intervals (CI), measuring the strength of the association of aggression and impulsivity (as continuous independent variables) with previous SB (as binary outcome measure), whilst adjusting for alcohol and drugs abuse, none of which were significantly associated with previous SB, thus testing hypothesis iii. As noted above (see hypothesis iii), we did not intend to statistically compare the strength of these associations across sites, which requires a different methodology [29]. Rather, we explored whether or not these associations were statistically significant for each site.

A significance level of 5% (two-tailed) was set for all the above analyses, performed using the Statistical Package for Social Science version 25.0 (SPSS, Inc., Chicago, IL, USA).

## 3. Results

### 3.1. Characteristics of the Samples

The demographic and clinical characteristics of the samples are shown in Table 1, including between-site differences.

### 3.2. Differences in Aggression and Impulsivity Levels across Sites

As detailed in Table 2 there were relevant between-site differences in aggression and impulsivity levels. In terms of aggression, NYC subjects (20.2 ± 5.6) had significantly (*p* < 0.001) higher mean BGS total scores than participants from Florence (16.1 ± 4.9) who had significantly (*p* = 0.022) greater BGS scores than those from Madrid (14.4 ± 5.3).

With regard to impulsivity, the mean BIS score was higher in the Florence sample (55.0 ± 9.7) than in Madrid (53.2 ± 20.1) or NYC (52.0 ± 16.6), although these differences were non-significant.

### 3.3. Differences in Aggression and Impulsivity Levels between Suicide Attempters and Non-Attempters

Suicide attempters showed higher BGS scores than non-attempters across sites (NYC: 22.0 ± 5.8 vs. 18.6 ± 4.9, *t* = 6.45, *p* < 0.001; Madrid: 14.9 ± 5.6 vs. 13.2 ± 4.3, *t* = 3.17, *p* = 0.002), while in Florence these differences showed a trend *p* (17.3 ± 5.3 vs. 15.3 ± 4.5, *t* = 1.86, *p* = 0.067), which is shown in Table 3.

Suicide attempters also had significantly higher BIS impulsivity scores than non-attempters in NYC (54.2 ± 16.6 vs. 50.1 ± 16.3, *t* = 2.72, *p* = 0.007) and in Madrid (56.3 ± 18.4 vs. 41.4 ± 21.6, *t* = 5.71, *p* < 0.001), although these differences were not observed in the Florence sample, as detailed in Table 3.

In keeping with the above, we have more graphically illustrated the above results in two figures. Thus, Figure 1 below, presents the BGS differences between suicide attempters and non-attempters across sites, and Figure 2 below, shows the BIS differences between suicide attempters and non-attempters across sites.

### 3.4. Association of Previous Suicidal Behaviour with Aggression and Impulsivity across Sites

Previous SB was significantly associated with aggression in NYC and in Florence as follows: NYC (OR 1.12, 95% CI 1.07–1.16; *p* < 0.001) and Florence (OR 1.11, 95% CI 1.01–1.22; *p* = 0.032). In Madrid, this association was nonsignificant (OR 1.05, 95% CI 0.99–1.11; *p* = 0.055).

The relationship between impulsivity and SB reached statistical significance in NYC (OR 1.01, 95% CI 1.00–1.02; *p* < 0.001) and in Madrid (OR 1.03, 95% CI 1.02–1.05; *p* < 0.001), while in Florence it was non-significant (OR 1.07, 95% CI 0.98–1.17; *p* = 0.12).

Table 4, below, presents the binary logistic regression models on previous SB. In bold, see the significant results.

## 4. Discussion

### 4.1. Main Findings

We compared three samples of convenience of patients with depressive disorders from NYC (US), Madrid (Spain) and Florence (Italy) to investigate between-site differences both in aggression and impulsivity levels and in the association of these two variables with SB.

In keeping with our first hypothesis based on our previous studies [18,19], NYC patients were found to have the highest levels of aggression, although these results were not replicated for impulsivity, with Florence subjects having higher impulsivity levels than those from NYC or Madrid, which partially conflicted with our hypothesis. As expected (hypothesis ii), suicide attempters had higher aggression levels than non-attempters in NYC and in Florence, although data from the Madrid sample did not support this. Consistent with hypothesis ii, impulsivity levels also distinguished suicide attempters from non-attempters in NYC and in Madrid, although the Florence sample did not replicate this. Hence, results supported hypothesis iii since only in the NYC sample both aggression and impulsivity were significantly associated with previous SB in the multivariable regression model, which was not replicated by the Madrid and Florence samples, in which only impulsivity (in Madrid) or aggression (in Florence) remained significantly linked with SB. These differences in the contribution of aggression and impulsivity to SB may explain, in part, the higher suicide rates in NYC in comparison with Madrid and Florence, which is discussed further below.

### 4.2. Aggression, Impulsivity and Suicidal Behaviour: Differences across Sites

Previous literature considered aggression to be a personality trait [30]. Consistent with this, there seems to be a link between Cluster B personality disorders, alcohol/substances-related problems and aggression [31,32] and we found some evidence of this. In particular, a very busy inner-city area, such as NYC, was linked with increased aggression levels and a significant association of previous SB with aggression. Of note, the highest proportion of people with alcohol/drugs use disorder was found in NYC (compared with Madrid and Florence). On the other hand, being religious is a well-documented protective factor for SB [33], although some studies failed to replicate this (e.g., [34]). Religion may have, therefore, mitigated the association between aggression or impulsivity and SB in the Florence cohort, in which 80% of participants reported to have a religion (which was significantly higher than in Madrid and NYC cohorts), despite the higher impulsivity levels of Florence subjects.

Impulsivity has been established as a risk factor for SB both in adults and in children [35], and we showed impulsivity levels to distinguish suicide attempters from non-attempters in NYC and in Madrid. Impulsivity is a common feature of several psychiatric disorders [36], such as bipolar disorder (BD) [37] and Cluster B personality disorders [38], particularly borderline personality disorder [39,40]. Moreover, motor impulsivity was reported to differentiate multiple- from single suicide attempters [41]. However, our study did not assess early life and recent stressful life events, which appear to mediate the relationship between impulsivity and SB [42,43], in line with the so-called Interpersonal Theory of Suicide (IPTS) [44]. Indeed, childhood abuse contributes to impulsivity and SB [45], hence acting as a potential mediator/confounder in this relationship. Further studies assessing these other factors interacting with impulsivity in leading to SB are needed.

Of relevance is the fact that we have replicated the association of SB with aggression and impulsivity in the NYC sample [46], while in Florence SB was only linked with aggression and in Madrid only impulsivity was related to SB, which may have been due to insufficient statistical power. However, relatively, in contrast to our results, one study revealed impulsivity to play a more relevant role in distinguishing suicide attempters from non-attempters than aggression in both bipolar depression and major depressive disorder [47]. This study therefore adds to previous literature since our results showed that only in NYC both aggression and impulsivity remained associated with SB, that is, that there seem to be relevant between-site differences in the association between aggression, impulsivity and SB, which should be taken into account when investigating a diathesis of SB across different cultural settings.

In summary, the findings from this study suggest that the association of aggression and impulsivity with SB appear to vary across countries. This may also explain between-site suicide rates differences. In fact, in 2016 NYC suicide rates were 8.40/100,000 [7], which was much higher than suicide rates for the same year in Madrid (4.72/100,000) [48]. In keeping with this, according to the World Health Organization [49] national suicide rates in 2016 were as follows: the US: 13.9/100,000 persons-year, Spain: 7.46/100,000 persons-year and Italy: 8.2/100,000 persons-year. It could, therefore, be envisioned that tackling aggression may successfully reduce the risk of suicide in those high aggression levels sites such as NYC, while aggression reducing measures in low aggression levels countries such as Spain or Italy may have less impact as a suicide prevention approach. On the other hand, our findings suggest that suicide prevention strategies in Madrid should focus on the management of impulsivity.

Specifically, both anti-convulsivants and selective-serotonin reuptake inhibitors (SSRIs) have shown to reduce both aggression and impulsivity levels [50], although further randomized controlled trials (RCT) looking at suicide-related outcomes are needed. In this regard, it is worth noting that SB is a common exclusion criterion in RCTs [51], which hampers this research area. Indeed, while there is strong evidence to support the efficacy of antidepressants for unipolar MDD [52], destabilizing risks appear to be unfavorable in bipolar depression [53]. Although RCTs tend to favor the use of antidepressants to prevent SB in unipolar depression, this effect may reflect a selection bias since participants in RCTs tend not to have a suicidal history [51]. Moreover, suicidal ideation appears to predict poor response to antidepressants [17].

In addition, patients with personality disorders frequently develop comorbid depressive episodes in their lifetime and antidepressants were postulated to reduce aggression, impulsivity and SB risk. Although there is limited evidence to recommend the use of antidepressants for the management of impulsivity and depressed mood in personality disorders [54], PD patients with high risk of SB are more likely to get excluded from RCTs. Results also suggest that religiosity and other social support structures may mitigate effects of aggression and impulsivity in other countries, such as Italy, and social interventions may have a role in suicide prevention.

### 4.3. Strengths and Limitations

We compared three large samples of patients with depressive disorders from major cities in different countries, namely NYC (US), Madrid (Spain) and Florence (Italy), to investigate differences in the relationship between aggression, impulsivity and SB. The same instruments for assessing aggression and impulsivity, which were the co-primary outcome measures of the study, were used across the board. This approach is, therefore, likely to increase the validity of the study findings.

However, the study has several limitations. First, the recruitment methods, which were naturalistic and not initially designed to test this study’s hypotheses, differed across sites. Second, the Florence sample size (*n* = 93) was significantly smaller than the other two cohorts (NYC: *n* = 647; Madrid: *n* = 658), although this did not prevent us from finding the aforementioned between-sample differences. Third, the same scales were used to measure aggression and impulsivity, BGS and BIS, which is a self-report scale, respectively, although no inter-rater reliability measures were calculated for BGS. Fourth, the cross-sectional design of the study did not permit us to reach causality conclusions [55]. Finally, we acknowledge that the methodology used for testing hypothesis iii may be subject to criticism given relevant between-sample differences. This said, it is worth noting that we aimed to cross-compare whether or not aggression and impulsivity remained significantly associated with SB across sites. In other words, we did not intend to statistically compare the strength of these associations across samples, which would require a different approach [29]. Truly, this is a different research question which remains to be addressed by future studies specifically designed for this.

### 4.4. Implications on Suicide Prevention and Future Research

Our results provide further support for incorporating aggression and impulsivity measure into routine clinical suicide risk assessments, although the association of these variables to SB appears to vary across different cultural settings, which is the main contribution of this work. In addition, it is worth noting that both aggression and impulsivity could be modified through intervention, including antidepressants, thus reducing the risk of SB [56]. Moreover, not only may early detection and intervention of aggression and impulsivity reduce the risk of SB, but so may the medical lethality of SBs when they occur.

Nevertheless, further replication cross-cultural studies comparing aggression and impulsivity levels in suicide attempters and non-attempters in other countries, including different settings (for instance, rural and urban areas, in- and outpatient settings), are warranted. Some methodological issues, such as the use of the same instruments, similar recruitment procedures and some reliability measures, need to be considered. In addition, bilingual assessors receiving common training at the different sites may increase the inter-rater reliability, which should be reported. Finally, it should be noted that funding of national healthcare systems differs across countries, from fully-publicly-funded healthcare systems, such as Spain and Italy, to private medical insurance-based countries such as the US. In addition, this study’s findings are unlikely to generalize to low- and middle-income countries, in which not only up to 79% of worldwide suicides occur, but also suicide research tends to be underfunded [57].

From a neurobiological research perspective, these cross-cultural differences should be considered by future studies looking at the biological underpinnings of aggression, impulsivity and SB. In particular, two previous neuroimaging studies from the US revealed some neuroanatomical correlations of impulsivity, aggression and SB. First, by using structural magnetic resonance imaging (MRI) in a sample of (*n* = 51) BPD suicide attempters from Pittsburgh (US), medical lethality of previous suicide acts was found to inversely correlate with grey matter volumes across multiple fronto-temporal-limbic regions [58]. Second, a recent small study of BD patients using diffusion tensor imaging revealed white matter changes in the anterior limb of the internal capsule and anterior corona radiate to play a role in impulsivity in suicide attempters in comparison with non-attempters [59]. Therefore, it seems that, irrespective of diagnosis, impulsivity-aggression may represent an endophenotype which significantly increases the risk of SB. The combination of machine-learning techniques [60] with cutting-edge neurobiological research methods may contribute to a better understanding of these complex associations. Furthermore, a deeper knowledge of the neurobiological underpinnings of aggression, impulsivity and SB, and their exacerbation in relationship with recent and remote trauma exposure, and different cultural contexts, may pave the way towards more targeted pharmacological treatments for the prevention and management of SB.

## 5. Conclusions

To sum up, this investigation revealed relevant cross-cultural differences in the contribution of aggression and impulsivity to SB, which are likely to have implications on the prevention and management of suicide, undoubtedly a pending global challenge, particularly in the post-COVID-19 years to come.

## Figures and Tables

**Figure 1 jcm-10-03057-f001:**
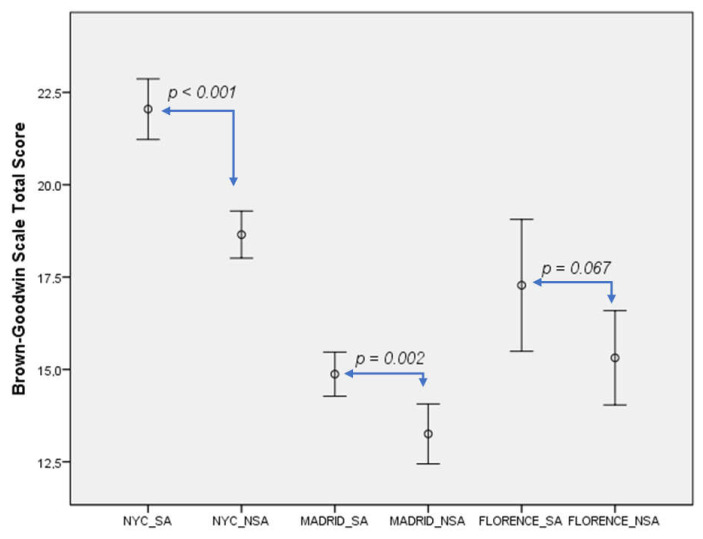
Aggression levels differences between suicide attempters and non-attempters across sites. NYC: New York City. SA: Suicide-attempters. NSA: non-suicide attempters.

**Figure 2 jcm-10-03057-f002:**
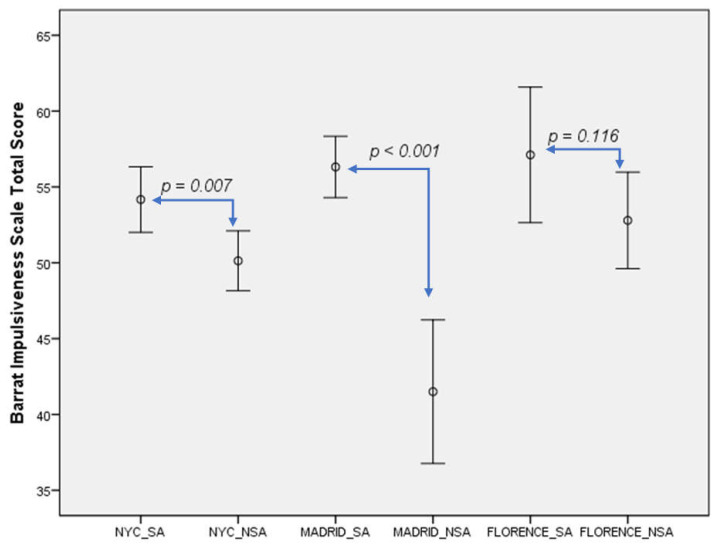
Impulsivity levels differences between suicide attempters and non-attempters across sites. NYC: New York City. SA: Suicide attempters. NSA: non-suicide attempter.

**Table 1 jcm-10-03057-t001:** Characteristics of the samples.

	NYC (*n* = 647)	MADRID (*n* = 658)	FLORENCE (*n* = 93)	*p*-Value
	Mean ± SD *n* (%)	Mean ± SD *n* (%)	Mean ± SD *n* (%)	
Age	37.8 ± 13.0	41.1 ± 15.1	47.3 ± 14.1	<0.001
Gender (males)	266 (41.1)	229 (34.8)	33 (35.5)	0.57
Education (high)	301 (48.2)	162 (27.2)	5 (14.3)	<0.001
Unmarried	491 (76.2)	400 (65.1)	50 (53.8)	<0.001
Unemployed	388 (60.7)	411 (62.5)	32 (43.2)	0.006
Children	245 (40.0)	274 (50.4)	44 (47.3)	0.002
Religion (vs. none)	485 (77.6)	187 (66.8)	52 (80.0)	<0.001
MDD	482 (74.5)	516 (78.4)	45 (48.4)	<0.001
BD	165 (25.5)	149 (22.6)	48 (51.6)	<0.001
Alcohol abuse	210 (32.5)	126 (23.8)	1 (1.0)	<0.001
Drugs abuse	160 (24.8)	93 (17.5)	7 (7.6)	<0.001
Childhood abuse	191 (34.3)	117 (23.5)	5 (5.4)	<0.001
Abuse lifetime	242 (43.9)	173 (34.8)	6 (6.4)	<0.001
Suicide attempters	303 (46.8)	465 (74.9)	39 (41.9)	<0.001

MDD: Major Depressive Disorder. BD: Bipolar Disorder.

**Table 2 jcm-10-03057-t002:** Differences in aggression and impulsivity scores across sites.

		NYC	MADRID	FLORENCE	Statistic	*p*-Value
BGS score		*n* = 441	*n* = 465	*n* = 87		
	Mean ± SD, 95% CI	20.2 ± 5.6, 19.6–20.7	14.4 ± 5.3, 13.9–14.9	16.1 ± 4.9, 15.1–17.2	F = 12.62	NYC > M, *p* < 0.001 NYC > F, *p* < 0.001 F > M, *p* = 0.022
BIS score		*n* = 494	*n* = 251	*n* = 50		
	Mean ± SD, 95% CI	52.0 ± 16.6, 50.5–53.5	53.2 ± 20.1, 51.2–55.1	55.0 ± 9.7, 52.3–57.8	F = 0.932	F > M, *p* = 0.766 F > NYC, *p* = 0.490 M > NYC, *p* = 0.598

BGS: Brown-Goodwin Scale [24]. BIS: Barratt Impulsiveness Scale [25]. NYC: New York City. M: Madrid. F: Florence.

**Table 3 jcm-10-03057-t003:** Differences in BGS and BIS total scores between suicide attempters and non-attempters across sites.

	NYC				MADRID				FLORENCE			
	Attempters	Non-Attempters	*t*	*p*-Value	Attempters	Non-Attempters	*t*	*p*-Value	Attempters	Non-Attempters	*t*	*p*-Value
BGS	*n (%)*	*n (%)*			*n (%)*	*n (%)*			*n (%)*	*n (%)*		
	193 (45.2)	234 (54.8)			343 (75.7)	110 (24.3)			36 (41.4)	51 (58.6)		
	*Mean ± SD*	*Mean ± SD*			*Mean ± SD*	*Mean ± SD*			*Mean ± SD*	*Mean ± SD*		
	22.0 ± 5.8	18.6 ± 4.9	6.45	<0.001	14.9 ± 5.6	13.2 ± 4.3	3.17	0.002	17.3 ± 5.3	15.3 ± 4.5	1.86	0.067
	*95% CI Mean*	*95% CI Mean*			*95% CI Mean*	*95% CI Mean*			*95% CI Mean*	*95% CI Mean*		
	21.2–22.9	18.0–19.3			14.3–15.5	12.4–14.1			15.5–19.1	14.0–16.6		
BIS	*n (%)*	*n (%)*			*n (%)*	*n (%)*			*n (%)*	*n (%)*		
	231 (46.6)	265 (53.4)			321 (79.6)	82 (20.3)			26 (52.0)	24 (48.0)		
	*Mean ± SD*	*Mean ± SD*			*Mean ± SD*	*Mean ± SD*			*Mean ± SD*	*Mean ± SD*		
	54.2 ± 16.6	50.1 ± 16.3	2.72	0.007	56.3 ± 18.4	41.4 ± 21.6	5.71	<0.001	57.1 ± 11.0	52.8 ± 7.5	1.60	0.116
	*95% CI Mean*	*95% CI Mean*			*95% CI Mean*	*95% CI Mean*			*95% CI Mean*	*95% CI Mean*		
	52.0–56.3	48.1–52.1			54.3–58.3	36.8–46.2			52.6–61.6	49.6–56.0		

BGS: Brown-Goodwin Scale [24]. BIS: Barratt Impulsiveness Scale [25].

**Table 4 jcm-10-03057-t004:** Binary logistic regression on previous SB.

	NYC			MADRID			FLORENCE		
	OR	95% CI	*p*-Value	OR	95% CI	*p*-Value	OR	95% CI	*p*-Value
	*n = 427*			*n = 453*			*n = 87*		
Alcohol	1.26	0.78–2.04	0.343	1.46	0.79–2.67	0.222			
Drugs	1.37	0.80–2.34	0.248	1.25	0.61–2.53	0.543	0.19	0.02–2.22	1.86
BGS	**1.12**	**1.07–1.16**	**<0.001**	1.05	0.99–1.11	0.055	**1.11**	**1.01–1.22**	**0.032**
	*n = 496*			*n = 251*			*n = 50*		
Alcohol	1.41	0.93–2.13	0.104	1.49	0.72–3.07	0.280			
Drugs	1.26	0.67–2.81	0.124	1.41	0.57–3.50	0.461	0.17	0.02–1.81	0.343
BIS	**1.01**	**1.00–1.02**	**0.044**	**1.03**	**1.02–1.05**	**<0.001**	1.07	0.98–1.17	0.12

BGS: Brown-Goodwin Scale [24]. BIS: Barratt Impulsiveness Scale [25].

## Data Availability

Datasets are available upon request, provided data policy access is complied with.

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
