# Peer review of "Aggression, Impulsivity and Suicidal Behavior in Depressive Disorders: A Comparison Study between New York City (US), Madrid (Spain) and Florence (Italy)"

_jcm, 2021, doi:10.3390/jcm10143057_

Round 1

Reviewer 1 Report

I'm grateful to the reviewers for taking into account our suggested changes and I think this significantly improves the manuscript.

Author Response

Dear editor,

The reviewer 1 thanked the authors for taking into account the suggested changes which significantly improved the manuscript according to him/her. The reviewer 1 made no further comments.

Reviewer 2 Report

The authors should state clearly whether the present manuscript is in whole or in part a re-analysis of previously published data, since it appears that they have conducted very similar studies in the past.

The authors state as their first hypothesis that "NYC subjects will show higher  aggression and impulsivity levels than those from Madrid and Florence;" However, the background literature presented does not justify this hypothesis. The same goes for the third hypothesis.

The testing of hypothesis iii is not appropriate, as odds ratios and their associated confidence intervals and p-values are not comparable across models (including when models are similar in terms of their variables, but estimated on different samples). Further, it goes without saying that the confidence intervals will be narrower and the p-value lower, when estimating a model on 427 patients than when estimating a similar model on 87 patients (or even 50).

Table 4 should mention the adjustment variables - this could be in a note, but the table should be readable on its own.

Author Response

Dear editor,

The reviewer 2 made the following comments.

First, the reviewer 2 asked us to clarify whether the present manuscript was a re-analysis of previously published data.

While the present study builds on previous work from our group, as mentioned in the introduction, these two studies tested different hypotheses, which has been clarified further in the introduction section of the revised manuscript as follows:

This said, it is worth noting that these studies tested different hypotheses. First, the role of the promoter area of the serotonin transporter (17q11.1-q12) was investigated in relation to history of aggression and impulsivity in two samples of suicide attempters and healthy controls from NYC and Madrid, which partially overlapped with the samples used in the present study (Baca-García et al., 2004). The second study aimed to compare the association between lethality of suicide attempts and aggression levels in Madrid and in NYC with the aforementioned two samples (Baca-García et al., 2006). However, the present study aimed to compare aggression and impulsivity levels across three sites, namely NYC, Madrid and Florence, and the relationship between both variables and SB in three samples of ‘depressive disorders patients’, that is, we did not include healthy controls. Hence, the present study is not a partial or full re-analysis of previously published data.

Second, the reviewer 2 suggested to further justify hypothesis i and hypothesis iii based on previous literature. Accordingly, we have amended the last paragraph of the introduction as follows:

In particular, hypothesis i and iii were based on our aforementioned previous studies [18,19], which are supported further by previous epidemiological research on suicide rates differences across countries [20]. The association of aggression-impulsivity with SB (hypothesis ii) has been previously reported [9–11].

Third, reviewer 2 raised concerns about the appropriateness of the methods used for testing hypothesis iii since the models are based on different samples sizes.

It is true that the samples, which were not originally recruited for conducting this study are rather different in terms of baseline characteristics (Table 1), also significantly differed in sample size which may affect the analyses. One may consider a meta-analysis to be a more appropriate approach, the very low number of samples (i.e., studies), only three, and anticipated heterogeneity across these samples, led us to take a different approach. Nevertheless, we have acknowledged this in the limitations section of the revised manuscript as follows:

Finally, we acknowledge that the methodology used for testing hypothesis iii may be subject to criticism given relevant between-sample differences. Although a meta-analysis on three samples (i.e., three studies) may be considered to be more appropriate, the very low number of studies (3) and anticipated heterogeneity made us take a semi-quantitative approach to cross-comparing the strength of the association between aggression-impulsivity and SB across sites.

Fourth, the reviewer 2 suggested including the adjustment variables in Table 4, which has been amended in the revised manuscript accordingly.

Thank you very much for considering our submission for publication.

Yours sincerely,

Dr Lopez-Morinigo, on behalf of all the authors

Round 2

Reviewer 2 Report

The authors have not addressed my two main questions:

  1. Did you reanalyze published data, adding new data, or were all data for this study collected for the purpose of this study? By this I mean include already published data. One way to clarify this would be to give the dates during which data were collected.
  2. The logistic coefficients cannot be compared across models. This is a somewhat technical criticism, and it appears that I did not make it sufficiently clear what is meant by this. I found a blog that does a better job than I can explaining the reasons, and I refer the authors to this blog: https://francish.netlify.app/post/comparing-coefficients-across-logistic-regression-models/ I have also attached a presentation that I found online that explains it well.

Author Response

7th June 2021

Dear Editor,

Thank you for your email of 02/06/2021 with regard to our submission: ‘jcm-1232589 – Aggression, impulsivity and suicidal behaviour in depressive disorders: a comparison study between New York City (US), Madrid (Spain) and Florence (Italy)’.

Firstly, we much appreciate your consideration of our manuscript for publication in Journal of Clinical Medicine. In addition, we are very grateful for the reviewers’ comments as they have helped us to improve the quality of the article.

In particular, regarding such comments we have introduced the following amendments as tracked changes in a revised manuscript which is enclosed, thus in compliance with the journal guidelines. These responses are also available below and also, we would like to highlight the following clarifications.

Reviewer 1:

            The reviewer 1 made no further comments to be addressed.

Reviewer 2:

The reviewer made two comments.

First, the reviewer asked us to clarify whether the present manuscript was a re-analysis of previously published data, whether we added new data or whether all data for this study specifically collected with this purpose. Specifically, the reviewer suggested providing the dates during which data were collected to clarify this.

While the present study builds on previous work from our group, as mentioned in the introduction, these two studies tested different hypotheses, which was clarified further in the introduction section of the revised manuscript. In addition, in the revised manuscript, which is enclosed, we have made the following clarification point (in the introduction section):

Two previous studies from our group, which compared two samples of suicide attempters with non-attempters major depressive disorder (MDD) patients and healthy controls from New York City (NYC) and Madrid (Spain) [18,19], showed that the greater lethality of SB in NYC subjects was related to their higher aggression levels in comparison with Madrid participants, while no differences in impulsivity scores between suicide attempters and healthy controls were revealed by the analyses. This said, it is worth noting that these studies tested different hypotheses. First, the role of the promoter area of the serotonin transporter (17q11.1-q12) was investigated in relation to history of aggression and impulsivity in two samples of suicide attempters and healthy controls from NYC and Madrid, which partially overlapped with the samples used in the present study [19]. The second study aimed to compare the association between lethality of suicide attempts and aggression levels in Madrid and in NYC with the aforementioned two samples [18]. However, the present study aimed to compare aggression and impulsivity levels across three sites, namely NYC, Madrid and Florence, and the relationship between these two variables and SB in three samples of ‘depressive disorders patients’, that is, we did not include healthy controls. Hence, the present study is not a partial or full re-analysis of previously published data. More specifically, unlike the aforementioned two reports [18,19], not only new data were collected in NYC and in Madrid, but also we included a third cohort of depressive disorders patients from Florence in order to test this study hypotheses. However, as acknowledged in the methods and limitations sections, these three samples were not originally recruited for this study. Rather, based on an international collaboration research network, the three groups decided to share the data, thus allowing this comparison study (also, see recruitment dates below).

Second, the reviewer stated that ‘The logistic coefficients cannot be compared across models’.

We totally agree with the reviewer on this point, although the study did not aim to statistically compare logistic coefficients across sites. Accordingly, we have amended both the introduction and statistics sections in order to clarify this further. Moreover, we have reformulated the conclusions drawn from results in section 4.1.-Main findings of the discussion- as follows:

            In the introduction section:

In particular, the following hypotheses were to be tested: i) that NYC subjects will show higher aggression and impulsivity levels than those from Madrid and Florence; ii) that suicide attempters will have higher aggression and impulsivity levels than non-attempters across sites, iii) that while both aggression and impulsivity will be significantly associated with previous SB in NYC, these associations will not reach significance in Madrid or in Florence.

In the statistics section:

As noted above (see hypothesis iii), we did not intend to statistically compare the strength of these associations across sites, which requires a different methodology [29]. Rather, we explored whether or not these associations were statistically significant for each site.

In the discussion (section 4.1., lines 224-239):

In keeping with our first hypothesis based on our previous studies [18,19], NYC patients were found to have the highest levels of aggression, although these results were not replicated for impulsivity, with Florence subjects having higher impulsivity levels than those from NYC or Madrid, which partially conflicted with our hypothesis. As expected (hypothesis ii), suicide attempters had higher aggression levels than non-attempters in NYC and in Florence, although data from the Madrid sample did not support this. Consistent with hypothesis ii, impulsivity levels also distinguished suicide attempters from non-attempters in NYC and in Madrid, although the Florence sample did not replicate this. Hence, results supported hypothesis iii since only in the NYC sample both aggression and impulsivity were significantly associated with previous SB in the multivariable regression model, which was not replicated by the Madrid and Florence samples, in which only impulsivity (in Madrid) or aggression (in Florence) remained significantly linked with SB. This may also explain, in part, the higher suicide rates in NYC in comparison with Madrid and Florence, which is discussed further below.

In the discussion section (lines 279-290):

Of relevance, we have replicated the association of SB with aggression and impulsivity in the NYC sample [46], while in Florence SB was only linked with aggression and in Madrid only impulsivity was related to SB, which may have been due to insufficient statistical power. However, relatively in contrast to our results, one study revealed impulsivity to play a more relevant role in distinguishing suicide attempters from non-attempters than aggression in both bipolar depression and major depressive disorder [47]. This study therefore adds to previous literature since our results showed that only in NYC both aggression and impulsivity remained associated with SB, that is, that there seem to be relevant between-site differences in the association between aggression, impulsivity and SB, which should be taken into account when investigating a diathesis of SB across different cultural settings.

In the limitations section (lines 338-343):

Finally, we acknowledge that the methodology used for testing hypothesis iii may be subject to criticism given relevant between-sample differences. This said, it is worth noting that we aimed to cross-compare whether or not aggression and impulsivity remained significantly associated with SB across sites. In other words, we did not intend to statistically compare the strength of these associations across samples, which would require a different approach [29]. Truly, this is a different research question which remains to be addressed by future studies specifically designed for this.

Thank you very much for considering our submission for publication.

Yours sincerely,

Dr Lopez-Morinigo, on behalf of all the authors

This manuscript is a resubmission of an earlier submission. The following is a list of the peer review reports and author responses from that submission.

Round 1

Reviewer 1 Report

The paper uniquely assesses for cultural differences in the associations between aggression, impulsivity, and suicidal behavior. While this could offer an interesting and important contribution to the literature, there are several methodological issues with the paper that make it difficult to interpret any of the results. Namely, there is very little information given about the three samples. Given that the core of the study is comparing these samples it is critical that sufficient information is given to understand how these samples may differ, such as inclusion/exclusion criteria. Additionally, the study states that it is looking at associations among depressive disorders but no information is given in the methods to characterize this. Were all participants diagnosed with a depressive disorder? Which disorders? How were they measured? The methods section must be significantly revamped to provide interpretable results. Additional feedback is below.

Introduction

  • The US suicide statistics are from 2016. There are more recent ones available

Methods

  • P2, Ln73: The sentence describing Madrid participants is not a complete sentence
  • Please give additional details about Florence subjects
  • For all samples, please give the inclusion/exclusion criteria for enrolling participants. The first sentence infers that all patients had clinical levels of depression, however, the sample descriptions indicate that participants may have simply been individuals who were hospitalized following a suicide attempt. Given that these samples are being directly contrasted to one another, additional details are needed to understand whether any significant differences may be attributable to different methods or sample characteristics.
  • Given that authors are contrasting aggression and impulsivity between cultures, it would be helpful to understand whether the measures are valid within each language. Please provide additional information regarding the validity of measures or note if the measures have not been validated in all of the languages it was delivered in.
  • The authors included alcohol and substance abuse, but no justification is given.
  • There is no measure of depression described, yet the paper states that it is assessing aggression, impulsivity and suicidal behavior in depressive disorders. It is unclear how depressive disorders plays a role in the study.
  • Similarly, it is unclear how suicidal behavior was conceptualized or measured.

Statistics

  • The authors state that hypothesis ii assesses for differences between attempters and non-attempters, but the methods section gives no information about how non-attempters were recruited.

Results

  • The table includes Bipolar Disorder, but this has not been mentioned anywhere before in the paper
  • The figures are illegible

Reviewer 2 Report

In this cross-sectional study, the authors aim to compare aggression and impulsivity levels between New York City, Madrid and Florence, as well as investigating the between-site differences in the association of aggression and impulsivity with previous suicidal behaviour. A strength of the paper is the use of the same instruments (the Brown-Goodwin Scale and the Barratt Impulsivity Scale) across international sites. The main finding of interest is the higher levels of impulsivity and aggression in people with previous suicidal behaviour across New York and Madrid. However, the current paper has significant limitations meaning that its current conclusions are not robustly supported by the data.

  1. A major limitation of this paper is the lack of adjustment for drug and alcohol use. Drug and alcohol use is a much more robust risk factor for suicide than impulsivity or aggression, and indeed drug and alcohol use is an important cause of impulsivity and aggression. Without adjusting for drug and alcohol use, the paper is misleading, so it is essential this is remedied in the multivariate analysis.
  2. The authors go on to discuss antidepressants as a means of improving aggression and impulsivity. Aggression and impulsivity are more commonly seen in personality disorder (or even bipolar disorder) than unipolar depression. Patients with personality disorder are more likely to benefit from psychological therapy, whilst antidepressants are usually cautioned in bipolar disorder. Moreover, antidepressants will have poor efficacy in dependent alcohol and drug users, emphasising again the importance of adjusting for alcohol and drug use in their analysis.
  3. The authors conclude, including in the abstract, “Aggression and impulsivity scores were higher in suicide attempters than in non-attempters across sites”. This isn’t true because findings for Florence were not significant.
  4. In the ROC analysis in section 3.2, it’s clear that these questionnaire scores are very poor predictors of suicidal behaviour (ROC values are not much better than 0.5, 0.5) so I suggest removing this section.
  5. The authors acknowledge the low sample size form Florence as a limitation but under-acknowledge the impact it has had on their findings. Many of the non-significant differences in the Florence sample are largely due to underpowering. See section 3.4 for example – the odds ratios for Florence are actually higher, just the sample far smaller.
  6. It doesn’t make sense to analyse suicidal behaviour as the predictor of impulsivity and aggression; clearly it should be the other way around. The authors are interested in preventing suicide, so suicidal behaviour is the strongest proxy measure of this outcome
  7. In sum, there is interest in this paper if the authors are more focussed on their analysis and can test whether impulsivity/aggression in suicide attempters is seen independently of drug and alcohol use.